# IoT Smart Flooring Supporting Active and Healthy Lifestyles

**DOI:** 10.3390/s23063162

**Published:** 2023-03-16

**Authors:** Federico Cocconcelli, Guido Matrella, Niccolò Mora, Ion Casu, David Alejandro Vargas Godoy, Paolo Ciampolini

**Affiliations:** Dipartimento di Ingegneria e Architettura, Università di Parma, Parco Area delle Scienze 181/A, 43124 Parma, Italy

**Keywords:** active and healthy ageing (A and HA), active assisted living (AAL), internet of things (IoT), smart floor, assistive technologies

## Abstract

The lack of physical exercise is among the most relevant factors in developing health issues, and strategies to incentivize active lifestyles are key to preventing these issues. The PLEINAIR project developed a framework for creating outdoor park equipment, exploiting the IoT paradigm to build “Outdoor Smart Objects” (OSO) for making physical activity more appealing and rewarding to a broad range of users, regardless of their age and fitness. This paper presents the design and implementation of a prominent demonstrator of the OSO concept, consisting of a smart, sensitive flooring, based on anti-trauma floors commonly found in kids playgrounds. The floor is equipped with pressure sensors (piezoresistors) and visual feedback (LED-strips), to offer an enhanced, interactive and personalized user experience. OSOs exploit distributed intelligence and are connected to the Cloud infrastructure by using a MQTT protocol; apps have then been developed for interacting with the PLEINAIR system. Although simple in its general concept, several challenges must be faced, related to the application range (which called for high pressure sensitivity) and the scalability of the approach (requiring to implement a hierarchical system architecture). Some prototypes were fabricated and tested in a public environment, providing positive feedback to both the technical design and the concept validation.

## 1. Introduction

According to the World Health Organization [1], physical activity (PA) is a key factor for health and well-being, contributing to preventing and managing widespread diseases such as cardiovascular diseases, diabetes, cancer, and improving mental health as well. Adequate PA is needed at all ages: the WHO report stresses how 25% of the global population does not meet recommended PA levels, with such a figure raising to 80% for the adolescent population. Similarly, PA may benefit older adults in many ways, allowing to preserve strength, balance, and mental skills.

Investments in policies supporting PA are generally regarded as one of the best practices in public health management, as far as cost-effectiveness and quality of life are concerned. Many factors, however, jeopardize better and wider diffusion of physical activity practices. Among them, PA is often perceived as being boring, too demanding, and possibly unsafe.

Internet of Things (IoT) technologies can be profitably exploited to overcome such limitations, making the user experience more rewarding, safe, and effective. IoT makes adaption, personalization, and guidance of PA routines much easier, allows for continuous health monitoring and for connecting to healthcare systems, and makes it possible to create fun and social interactions as well.

IoT is widely exploited for health-oriented applications [2,3,4,5], including highly specific solutions, supporting the needs of people with disabilities [6,7], or techniques for non-intrusive, continuous monitoring of relevant physiological parameters (e.g., blood pressure [8], heart rate [9]). Wearable sensors are widely used, with the non-wearable approach being sought for further reducing the burden of the user [10,11]. The adoption of Artificial Intelligence techniques allows for indirect health assessment through activity recognition and behavioural analysis [12,13,14,15,16].

This paper discusses some results of the PLEINAIR project [17] (Parchi Liberi E Inclusivi in Network per Attività Intergenerazionale Ricreativa e fisica-Free And Inclusive Parks in Network for Recreational and Physical Intergenerational Activity), funded by Regione Emilia-Romagna, Italy. The project aims at promoting physical activity in outdoor environments by introducing smart equipment and by enriching the user experience. The project targets people of any age and fitness, with an inclusive approach.

More specifically, details on the implementation of the supporting infrastructure developed in the project are provided below. The main concepts of PLEINAIR were actually developed and refined through a comprehensive human-centred design assessment [18]: stakeholders and potential users were involved in all design phases, from the early feature specification to the final demonstrator testing. The PLEINAIR system components include: a set of “smart” tools, supporting different kinds of physical activities, a cloud environment to gather and process usage data, and interfaces for user interaction. Smart park gear relies on the so-called Outdoor Smart Objects (OSO) concept, i.e., a piece of park equipment enriched with sensing, processing and communication capabilities (thus adhering to the IoT paradigm). All OSOs feature a “common part”, accounting for communication and back-office management, and a more specific part, accounting for (two-way) user interaction. Although the OSO concept is much wider, in the following we shall focus on the development of the primary demonstrator developed within the project: a pressure-sensitive smart tile, consisting of a layer of rubber anti-trauma material (of the kind frequently found in kids playgrounds) embedding sensors and a visual interface (based on individually addressable, multicolor LEDs). Such a tile was exploited to implement a set of various exercises (games). Besides the tile itself, a general-purpose architecture has been devised, which is suitable for supporting further kinds of OSOs and for scaling up the aimed application.

## 2. Materials and Methods

### 2.1. Related Works

Smart flooring solutions have been devised for different purposes for quite a long time. In different clinical contexts, sensorized mats are commonly exploited for gait analysis. With reference to activity recognition and behavioural analysis, studies for identifying people by their footstep profiles were proposed in Refs. [19,20]. More recently, in Ref. [21], the ability to discriminate human footsteps from other events by analysing the output of a piezoelectric polymer floor sensor was demonstrated. Further recent works on the application of smart flooring to indoor localization can be found in Ref. [22]. In Ref. [23], Chang et al. exploit piezoresistive sensors instead as inputs for their Ubi-floor. In their study, a smart interactive flooring is built by deploying a dedicated electro-mechanic structure, embedding loudspeakers, and lighting, which can provide multimedia content and games. Some problems related to access and security management in smart parks are addressed in Ref. [24]. Smart flooring is used for localization and fall detection services, watching over elderly people at home. Capacitive sensors are used to this purpose in Refs. [25,26], among many others. A smart flooring equipped with capacitive sensors is used to provide localization and to identify fall detection. When the deployment of new smart flooring is unpractical or too expensive, smart carpets can be used instead (e.g., Ref. [27]).

In an Ambient Assisted Living (AAL) scenario, indoor localization is a key feature, but it needs to deal with privacy issues and acceptance by the users. Smart flooring allows for such a purpose without requiring the adoption of intrusive video technologies [28,29,30]. To deal with power consumption constraints, the triboelectric effect can be leveraged as well: triboelectrification, which is increasingly exploited for energy harvesting [31], has been shown to be suitable for footstep recognition and fall detection [32,33] and indoor localization as well [34]. In Ref. [35], a review of energy harvesting based on smart flooring is given.

Behavioural analysis, aimed at the recognition of daily living activities, enables a set of outcomes relevant to health assessment [36]: monitoring changes in daily habits patterns (either abrupt or slow, possibly related to health issues), support early detection and prevention in many cases. Smart flooring may contribute to the overall behavioural picture, and can be integrated into a multi-sensor environment. For instance, in Ref. [37] the sit-to-stand transfer duration is monitored, by combining sensitive mats on the floor and on the bed. In Ref. [38], instead, an activity dataset is provided, aimed at training and comparing different approaches for activity recognition and classification [39,40].

Even though many kinds of smart flooring technologies and applications are found in the literature, the peculiarities of the PLEINAIR application call for a novel implementation. The required features include:The adoption of the standard anti-trauma layer as the floor surface is needed to ensure safety while practising PA. Anti-trauma rubber (usually obtained by recycling exhausted car tires) is quite a rough material, and is not very suitable for accurate sensing. As illustrated below, a multi-layer solution has been adopted. Nevertheless, the unevenness of such a material makes its characterization and the a priori tuning of the read-out circuitry difficult. Therefore, an adaptive, self-calibrating scheme is needed, which is introduced below;The PLEINAIR flooring behaviour is inherently bi-directional, with lighting embedded in the tiles to display game guidance patterns. Both input and output signals are then to be managed by the control unit;A modular approach is to be implemented, to build arbitrarily sized flooring simply by assembling basic tiles;A cloud-based architecture is exploited for data collection and interfacing to healthcare systems, caregivers, and users through different fashions. However, cloud interaction is subject to timing and network availability issues, which makes it unpractical to manage full control of smart tiles at the cloud level. Hence, a hierarchical approach is to be followed, with system intelligence distributed at different levels: a local control unit, needed for managing cloud communication, is in charge of fast response tasks, related for instance to the game execution. A modular, scalable architecture has been devised, which is described in the following sections;The pressure-sensitive floor tiles lend themselves to further different applications, not necessarily related to walking and footsteps. A comprehensive review of aimed applications was carried out by the project team, including bio-mechanical engineers and physiotherapists, to set design specifications: among them, demanding constraints were assumed in terms of spatial resolution, time response, and pressure sensitivity. In particular, an activation threshold of 300 g/tile was assumed: even though it largely exceeds the needs for footstep recognition, such high sensitivity is required to enable different applications. For instance, the sensitive tile was used for implementing games to improve hand coordination skills, aimed primarily at people with disabilities. Similarly, an optimal time resolution of 10 ms was specified, to ensure adequate game promptness;The aimed outdoor deployment introduces a set of additional constraints, in terms of robustness, installation techniques, waterproofness, power sourcing, etc. The engineering phase, strictly related to device production, is still to be completed, however. For the sake of conciseness, the technicalities related to such constraints are not fully discussed in the following.

Based on this set of specifications, the design of PLEINAIR smart flooring was carried out.

### 2.2. The Smart Flooring System

The PLEINAIR project aims at stimulating an active lifestyle, triggering user engagement using a gaming, interactive approach. The smart tiles introduced above allows for several game patterns to be implemented: for instance, the classical hopscotch game can be straightforwardly implemented and enriched with dynamic colour lighting patterns, online performance comparisons, contests among friends, etc. By assuming a large enough surface covered by connected tiles, many further games can be implemented: a comprehensive description of gaming strategy and opportunities exceeds the scope of this technical paper and will be discussed elsewhere; here we shall focus mostly on the design, implementation and test of the smart floor IoT system.

For the sake of simplicity, let us think of an elementary game, in which the user has to walk by following the path indicated by lighted tiles. Taking advantage of the embedded intelligence and of the cloud connection, such a simple game can be highly personalized, accounting for the user’s age, skill, previous performances and medical recommendations. Parameters such as game speed, time length, and path complexity (jumps, different step lengths) can be tuned accordingly, based on the cloud-stored user’s profile.

Despite its simplicity, the above description allows us to highlight the need to coordinate the game action over several tiles, exploiting a scalable control strategy. The overall system architecture is hence summarized in Figure 1, and consists of:Anti-trauma tiles (brown squares);Embedded piezoresistive sensors (yellow circles);LED strips (light blue square lines);Tile Remote Input/Output boards (RIO, red boxes);Game controller and Wi-Fi communication unit (control, blue box);User app, connecting to the cloud. Different apps can be envisaged, supporting different roles, i.e., park user, caregivers, GP’s, healthcare system, etc.

**Figure 1 sensors-23-03162-f001:**
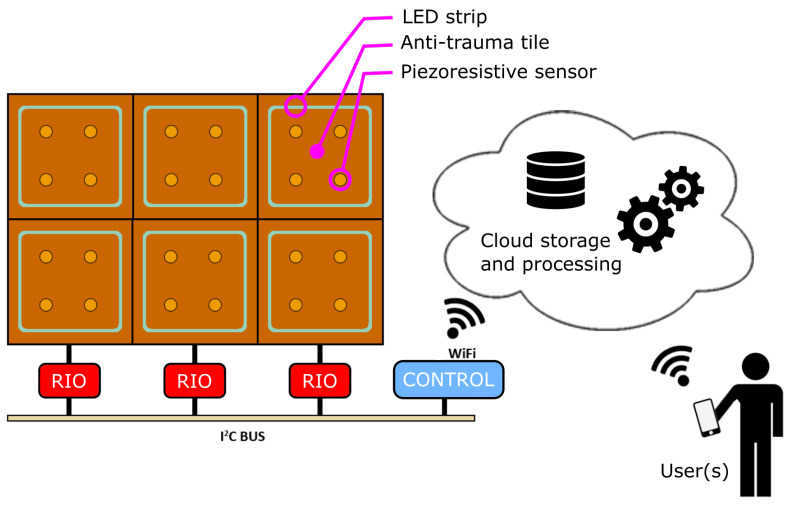
Schematic view of the smart flooring system.

#### 2.2.1. Smart Tiles

Tiles are made of recycled rubber and come with different thicknesses: on the example at hand, a 5 cm thickness was considered. Our tiles feature a square geometry, with a 50 cm edge and come in 2-tile mats (i.e., 100 cm × 50 cm). In the example mentioned below, a total surface of 8 square meters was covered (200 cm × 400 cm), requiring 32 tiles (16 double-tile mats). Each tile embeds an array of 40 WS2812B (WorldSemi Co., Ltd., Dongguan, Guangdong, China) addressable, multicolour LEDs. A single RIO board (I/O controller) can be shared by the two tiles in the same mat, and feature a MicroController Unit (MCU) for local processing and self-calibration of the sensor circuit. To address the 10 ms time resolution specification in a scalable fashion, RIO boards communicate with the game controller through an I^2^C bus.

In the following sections, details are given on the mechanical structure (Section 2.2.2), the electronic system (Section 2.2.3), the communication strategy (Section 2.2.4), and the game algorithm management (Section 2.2.5).

#### 2.2.2. Mechanical Structure

The smart tile is assembled by overlapping different layers: as mentioned above, the anti-trauma rubber roughness does not allow for precise sensor placement and reading, so a PVC layer is placed between the rubber and the sensors themselves, to more evenly distribute the pressure transmitted by the rubber layer. Similarly, a bottom PVC layer is placed below the sensors: outdoor deployment implies that the anti-trauma flooring should be deployed on a concrete slab, the surface of which can be quite rough as well. The bottom PVC layer serves both purposes of providing better levelling and insulation from ground moisture. A cross-sectional view of the smart tile is shown in Figure 2.

Four pressure sensors are embedded into each tile, possibly to allow for a spatial resolution finer than the tile pitch: the upper PVC layer is 3 mm thick, and it is flexible enough not to jeopardize such a spatial resolution, at the same time allowing for correctly spreading the force distribution above the sensors, regardless of the rough rubber texture. For a more stable and reliable assembly, each sensor is backed by a larger supporting washer.

Piezoelectric sensors were selected as sensing elements, after some comparative reasoning: in particular, weighting cells were discarded because of higher cost and of the need for more precise mechanical fixture. Piezoelectric sensors, instead, were discarded because of their inability to easily cope with static conditions (i.e., constant pressure). Piezoresistive sensors provide easier management of static conditions and do not require a critical fixture. Proper calibration and signal conditioning should be accounted for, however. The Flexiforce^TM^ A201 (Tekscan Inc., South Boston, MA, USA) piezoresistive sensor was selected, due to its small dimensions (the thickness is only 0.2 mm) and its large force range. The Flexiforce^TM^ A201 features a negative resistance coefficient, i.e., the resistance value decreases with the increasing force.

The optimal number of sensors per tile was experimentally determined (see Section 3.2 below), based on trading off cost, spatial resolution, and sensitivity. Although simpler applications may not require too fine a spatial resolution, having multiple sensors on each tile can help avoid any blind zones possibly caused by a non-uniform rubber texture.

Such a simple mechanical structure brings several advantages, in terms of cost, ease of installation, and compatibility with customary materials; however, non-uniform density of rubber material and loose installation techniques may result in an uneven distribution of the tile’s own weight; this requires individual calibration of the sensors, as introduced in Section 2.2.3.

#### 2.2.3. Electronic System

Besides the tiles themselves, the electronic system consists of the master board (game controller) and the RIO modules, connected through an I^2^C bus. The master board is a B-L475E-IOT01A Discovery kit from STMicroelectronics (STM S.p.a., Agrate Brianza, Italy), which mounts an STM32L4 series microcontroller based on Arm^®^ (ARM Ltd., Cambridge, UK) Cortex^®^ M4 processor and an ISM43362-M3G-L44 module by Inventek (Inventek System, Billerica, Massachusetts, USA) for Wi-Fi connectivity. Instead, the RIO board was specifically designed for this implementation. Figure 3 shows a block diagram of the RIO board. Eight sensors can be connected to each RIO module, allowing two tiles to be managed by a single module. A single sensor channel is shown in the figure. The sensor signal is first fed to a non-inverting amplifier stage, which accounts for pressure sensitivity, and is then filtered to remove the power line interference; a second-order Sallen–Key filter is used for this purpose. Then, a feedback loop is exploited for individual sensor dynamic calibration: at each system start-up, the sensor output is sampled by the MCU (through the embedded ADC) and a feedback signal is generated, suitable for zeroing the output of the differential amplifier (third stage). This process allows for the alignment of the sensors outputs, regardless of the different weight offsets. After the initial calibration phase, subsequent pressure on the tile results in a voltage increase at the related channels.

The RIO control logic exploits a NUCLEO-L432KC board from STMicroelectronics (STM S.p.a, Agrate Brianza, Italy), which embeds ADC converters and a 3.3 V LDO voltage regulator, which powers the external DAC and conditioning circuits. The LED strips (2 per RIO board) are independently powered at 5 V, and are controlled by the MCU through Pulse Width Modulation (PWM) signals, managed by on-strip WS2812B controllers by Worldsemi (WorldSemi Co., Ltd., Dongguan, Guangdong, China). An image of the implemented RIO board is shown in Figure 4.

Figure 5 illustrates the simulated sensor response to different pressures and different calibration voltages (i.e., the feedback voltage provided through the DAC converter). Increasing pressure results in a resistance decrease, as mentioned. Thanks to the conditioning chain, a marked threshold behaviour is shown, with the output voltage abruptly raising from 0 to 3.3 V (power voltage) as soon as the pressure increases. The feedback voltage that is worked out by the MCU and transferred through the DAC allows for tuning such a threshold, in order to deal with the different weight offsets at the different tiles. The closer the bias point is to the foot of the transition ramp, the lower change is needed in pressure/resistance to trigger the increase of the output voltage. The effect of the self-calibration procedure can thus be appreciated by comparing different curves in the figure, which differ in feedback voltage: such a feedback voltage can be used indeed to adjust the pressure threshold close to the actual value at rest. The threshold is programmable through the system software, allowing to balance sensitivity and noise immunity with reference to the specific situation at hand.

#### 2.2.4. Communication Infrastructure and Configuration Procedure

All RIO boards and the game controller board exchange information over the I^2^C bus. for this purpose, all boards feature an I^2^C bus extender (namely the P82B715 chip from NXP Semiconductors, Amsterdam, The Netherlands). The I^2^C protocol accounts for collision avoidance, allowing for scalability and reliability of the approach: each RIO board can take control of the bus as soon as a tile pressure change is detected, with the protocol effectively managing possible overlapping events.

In order to implement any game, the floor geometry must be known, and each tile needs to be properly identified by a unique ID and mapped onto the floor geometry. To allow for flexibility and scalability, the ID map cannot be hardwired or statically defined: a map “discovery” procedure has therefore been implemented, capable of dynamically building the ID map topology at run time. For this purpose, each RIO board features a push-button, to be used for flooring configuration. In the game description, the flooring topology is described by a 1D array of tiles; for practical reasons, however, tiles can be connected to the I^2^C bus in any order. At the game configuration, physical tiles should map to a set of consecutive ID’s: this is obtained simply by pushing board buttons in the topological sequence. At each push, the related board receives and permanently stores its ID, so that configuration is to be carried out only once for a given flooring geometry, regardless of subsequent changes in the game rules. This also allows for the quick self-test of the tiles network: at subsequent game startups, the game controller scans the ID array: should any RIO board fail to reply, the scan stops at the missing ID and a diagnostic pattern is shown through the tiles LED strips. Once a faulty board has been discovered, it can be replaced with no need for re-configuring the whole network from scratch. Once the replacement board has been connected to the bus, the discovery scan sequence is broadcast again: when it stops at the missing ID, the new board button is pushed, and the missing ID is stored in the replacement board memory. The discovery procedure is summarized by the flowchart shown in Figure 6.

In Figure 7, a view of the assembly of the smart flooring OSO deployed at the demonstrator site described in Section 4 below is shown, highlighting the distributed-intelligence architecture: tile-control boxes embed the RIO board and bus connections. The game controller (main board, connecting to the cloud) is not shown.

#### 2.2.5. Cloud Architecture and Game Execution

A more general view of the Pleinair IoT system architecture is shown in Figure 8. It consists of a Cloud Server, which provides services to all the different OSOs and to the PLEINAIR user app (developed in the project) (via API-rest). The system is based on hierarchically-distributed intelligence, aiming at better system scalability and responsiveness.

The basic building block of the system is the OSO: each OSO is an autonomous entity equipped with its own master microcontroller board (B-L475E-IOT01A, discussed in Section 2.2.3) and is connected to the Internet via Wi-Fi. A set of different OSOs can be referred to as a PLEINAIR smart park. In the case of the demonstrator discussed in Section 4 below, four OSOs were taken into account. Each OSO, in turn, may be based on modular architecture, featuring sensor clusters, as discussed in Section 2.2.4. Different parks can be accessed through the same PLEINAIR server and app: by doing so, the user is allowed to access different environments with the same credentials. At the same time, the user profile and personalization features can be shared and updated by different activities. In other words, the user management is dealt with at the cloud level, also opening up to the possibility of sharing data and devising integrated services in a wider smart-city or e-health scenarios. However, cloud communication is subject to network availability and limited speed, which can possibly jeopardize the system responsiveness and user experience while interacting with OSOs. To avoid this, exercise or game execution is managed by the local intelligence embedded in RIO boards, as introduced in Section 2.2.3 above.

At start-up, each OSO connects to the PLEINAIR Cloud Server and receives a certificate, to be compared with data stored in the firmware of its master board and to establish secure communication with the server. Each OSO features unique credentials (username and password). Similarly, users are authenticated when starting a session through the app, with their unique credential, to access their own personal profile. All cloud communication is encrypted through the TLS 1.2 (Transport Layer Security) protocol, to ensure data security.

After connection, the system self-configures itself, through the discovery procedure and the calibration phase described above. Then, the game controller eventually enters the gaming mode, waiting for a user request.

On the user’s side, after authentication, the user can select games and game controls through the web app; besides this, further information, such as performance, history, comparisons, and personalized guidance (based on activity records and on user’s health, age, and fitness parameters) can be shown.

Once the OSO of interest is selected, the user selects the game among a set of possible ones, possibly aiming at training different skills, and the challenge level.

The user’s options and commands are communicated to the PLEINAIR Server by API-rest functions and exploits the MQTT (Message Queue Telemetry Transport) protocol, which is widely used in IoT applications [41,42].

The server replies by sending game files and parameters to the OSO: once the game is started, game execution is fully managed on-board, with no need for cloud interaction; Computational resources (CPU, memory) of the MCU board are more than adequate: no meaningful constraints on game speed and complexity were actually encountered. At each game completion, a short report file in JSON format is uploaded to the cloud, to update the user’s statistics, with execution time, errors, etc. Each game report requires to transfer less than 2 KB of data, which makes the usage of the network resources quite limited.

Based on such statistics, information such as rankings, suggestions, and encouragement messages can be fed back to the user to boost his engagement and motivation.

Different games can be conceived, aiming at stimulating different PA features: speed, strength, coordination, balance, etc. Game design is a strictly interdisciplinary task, involving technical, bio-mechanical, and communication and sociological competencies: a discussion of the interdisciplinary fundamentals backing each game goes however beyond the scope of this paper.

Similarly, details about the web app implementation were managed by project partners and will be described elsewhere.

From the technical point of view, all games can be regarded, in principle, as a finite state machine, with inputs coming from sensors and the web user interface, and outputs consisting of LED-based displays and web app display.

In the demonstration site introduced below, different games (i.e., different FSM’s) were described by C-language coding and firmware programming. In a more general perspective, over-the-air (OTA) game download can be accounted for, allowing for different games to be played on the very same infrastructure with no need to physically access the controller board.

Such a hierarchical approach, which manages user profiles and game configuration at the cloud level and game execution at the local processing level, allows for great flexibility and scalability, with the cloud architecture being inherently suitable for serving differently sized domains, and the local architecture being independent of the overall system articulation.

## 3. Characterization and Test

### 3.1. Description of the Experimental Setup

Design specifications include the sensitivity of the smart tile to weigh as little as 300 g, placed at any position on the tile itself. Hence, the weight and position sensitivity was tested through a series of measures, in which a variable mass (a plastic container filled with sand) was placed at different positions on the tile.

The analogue output voltage and the resistances of the piezoresistive sensors were measured through an oscilloscope and a digital multimeter.

### 3.2. Experimental Results

Figure 9 shows the resistance variation with respect to the applied mass. The mass was placed over the tile and positioned above the sensor, then gradually increased, with steps of 100 g. The experimental results show that the resistance varies in an inverse proportion with force, as expected. For the example at hand, it is shown that the resistance shift for the minimum weight is in the order of some 20 kΩ, which is enough for reliable discrimination.

Then, the position sensitivity was evaluated, resulting from the layered structure described in Section 2.2.2 above. First, we considered what happens when pressure is applied in a region close to the sensor placement. For this purpose, we considered a quarter of the tile, with the sensor placed at the quarter centre and gradually increasing the weight being applied to different positions, numbered from 1 to 9, as shown in Figure 10a. The sensor was located under position 5. Figure 10b shows the sensor response: it is shown that, regardless of the weight placement, a notable swing of the analogue output voltage is obtained. The response distribution does not depend on the geometrical distance from the sensor only, but also depends on the different mechanical boundary constraints: e.g., although geometrically equidistant from the sensor, the response of square 3 (on the tile “free” edge) is much larger than that of square 7 (which connects to the remaining tile quarters).

Figure 11 replicates the same experiment on larger distances: in this case, the whole tile is considered, with sensors placed in quarter-centre positions. In this case, it is shown that for larger distances the sensor response is much lower, possibly not large enough for noise discrimination. In Figure, the response of Sensor 1 is shown, and pressure applied to peripheral positions (1, 2, 3, 4, 7) result in quite a low voltage swing. Similar results are obtained by sampling the remaining sensors. Likewise, if just a single sensor was placed in the tile centre, peripheral weighting would result in insufficient output voltage swing. Based on such results, we adopted the 4-sensor configuration, which yields an activation swing larger than 500 mV, regardless of the tile pressure point. Such a multiple sensor configuration, in principle, should allow for obtaining a spatial resolution finer than the tile pitch: in this case, however, the structural asymmetries mentioned above should be taken into account. Similarly, the rough texture of the rubber layer may jeopardize the accuracy of such analogue processing, calling for some calibration and training processes. In the present case, instead, binary information is to be extracted from the whole tile: from the figures, the conservative threshold of 500 mV voltage shift at any sensor can be assumed for assessing tile activation.

## 4. Discussion

A set of OSOs was implemented and deployed at a demonstrator site (Figure 12): a public place, the “Museo della Civiltà Contadina” in Bentivoglio (Bologna, Italy) was chosen, which allowed us to carry out tests with the general public and with selected users. A covered area was exploited in the entrance museum hall (a sort of veranda, closed by glass panels). Despite the aimed outdoor destination, deployment in a closed space was needed for safety purposes, with access to the PLEINAIR experimental facilities being constantly supervised.

Four IoT OSOs were installed at the demo site, mostly based on the smart-tile concept:**Smart flooring**: a 4 m × 2 m sensitive flooring, with 32 smart tiles covering the 8 m^2^ surface. The demo game required the user to follow a random path, indicated by the tiles lighting up in a sequence. If an out-of-sequence tile is stepped on, the user gets penalty points. The game is initiated by the web app (a screenshot, which is shown in Figure 13), and features different difficulty levels, based on the user’s profile. The overall score comes from the speed in completing the path and from the penalties for incorrect sequences;**Smart bench**: a sensorized stepper, exploiting two tiles. One tile is embedded in the seat, and the other one is on the pavement. A random sequence of activation is generated, with the user being asked to stand up and sitaccording to the colour code shown by the floor tile. The game stimulates strength, promptness, and coordination. It is controlled by the same web app: users were given a tablet to choose and start any of the available four games;**Smart table**: the table surface is made up of a couple of smart tiles, and the user is asked to press each tile with their hands when prompted by the colour code displayed by each tile. This game stimulates promptness and coordination, and was especially conceived for people with motor disabilities: the table structure was designed to allow for a wheelchair user to get close to the table surface. The high sensitivity of the tiles (as mentioned, a weight as small as 300 g can be detected) was fully exploited in this case, allowing people with weak upper limbs to enjoy the game;**Smart planter**: it is meant to demonstrate the flexibility and generality of the control architecture. In this case, no smart tiles were involved: the aim of the “game” was the remote control of a green plant, by monitoring environmental parameters such as temperature, humidity, atmospheric pressure, soil moisture, and allowing to manage additional lighting through a LED source. Despite the different purposes, the same general architecture and web app was used in this case too.

The testing phase lasted about three months, and allowed us to validate the full PLEINAIR ecosystem, including OSOs, the WebApp, and the cloud infrastructure. More than 200 users were involved in tests, and their reactions were assessed through interviews and questionnaires, a sample of which is shown in Figure 14.

Overall, the user satisfaction was fairly good and, from a technical point of view, no meaningful issues emerged during the testing period. Most relevant project choices were therefore validated:Despite its rough and non-homogeneous texture, the adoption of the standard anti-trauma rubber mats was eventually feasible, this contributing to keeping a familiar look and feel for the OSOs and perspectively allowing for deployment in many scenarios;By leveraging the gaming aspects, the goal of stimulating PA was attained; the test involved different class ages, ranging from primary school pupils to elderly association members. Through personalization, such users were able to find motivation and reward, regardless of their age and physical fitness;A general-purpose IoT architecture has been devised and implemented, suitable for scaling up and for a much wider scope than that involved in the demonstration.

This preliminary work opens up many possible developments and improvements, mostly in terms of more sophisticated and varied gaming procedures and better engagement of people with special needs.

## 5. Conclusions

In summary, the design and implementation of a set of smart equipment supporting healthy lifestyle and physical activity has been described in this study. The OSO concept complies with the IoT paradigm, taking full advantage of its flexibility and versatility. A smart tile has been developed by exploiting standard playground material, which lends itself to many different applications. Distributed tile-control modules were designed and implemented, allowing for scalable implementation. A cloud architecture has been implemented, dealing with safe and reliable communication: although the testing involved a single demonstration site, the cloud approach could easily extend to the management of a much wider set of OSOs, gathered at different “smart parks”. In addition, the usage data of such smart parks could be of great interest to social and health care systems, contributing to a big-data pool suitable for better assessing citizen’s needs and for interacting in a smart-city scenario. The PLEINAIR concept was validated through a demonstrative site, featuring four different OSOs that are open to the public: the interested reader is referred to a short video [43], which illustrates the most relevant project features. The outcomes of the testing phase were encouraging and showed that IoT technology can be profitably used in such a context as well, fostering the adoption of healthier and more active lifestyles.

## Figures and Tables

**Figure 2 sensors-23-03162-f002:**
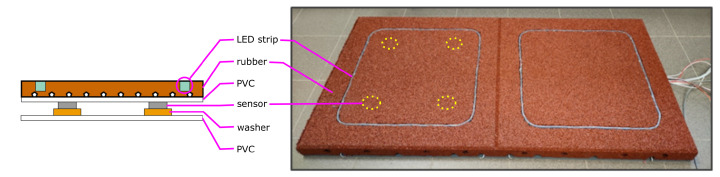
Cross-section of the smart tile (**left**) and an actual image of a two-tile mat (**right**).

**Figure 3 sensors-23-03162-f003:**
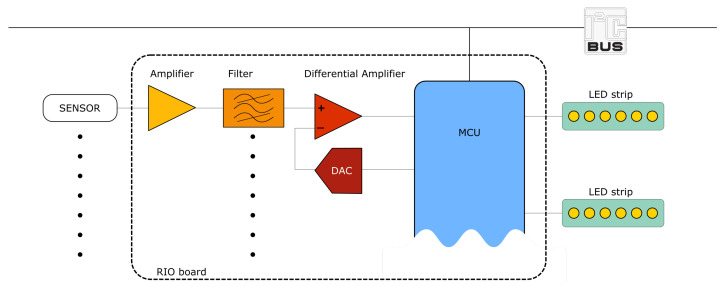
Block diagram of the RIO board.

**Figure 4 sensors-23-03162-f004:**
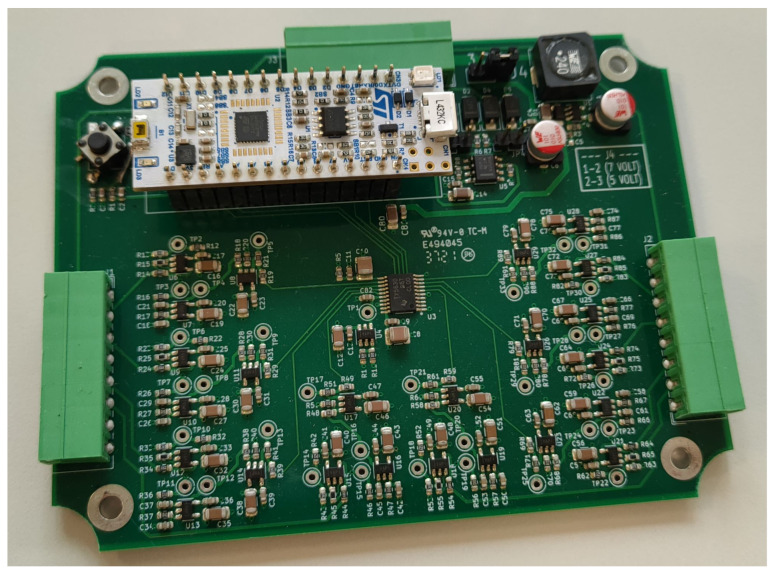
View of the realized RIO board.

**Figure 5 sensors-23-03162-f005:**
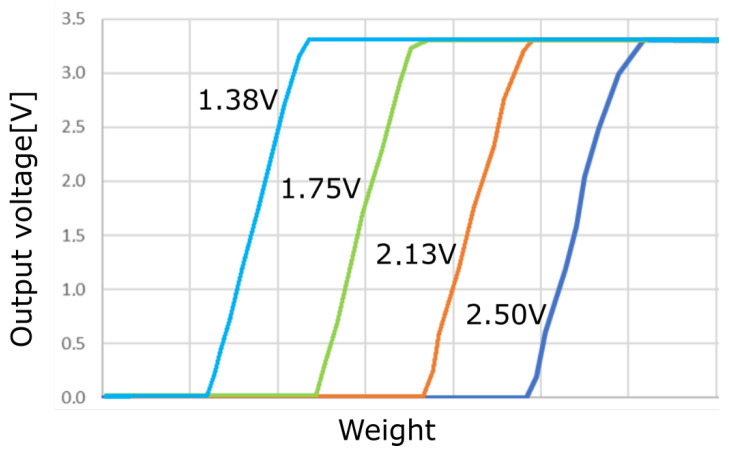
Simulation illustrating the calibrating procedure. Each curve is labelled by the related calibration voltage. The weight is swept over a meaningful range: however, the electrical simulation does not account for the actual pressure distribution through the layered mechanical structure but assumes a fixed conversion instead from weight to electrical conductance. For this reason, only a qualitative effect is shown here, with more quantitative details coming from the measurements below.

**Figure 6 sensors-23-03162-f006:**
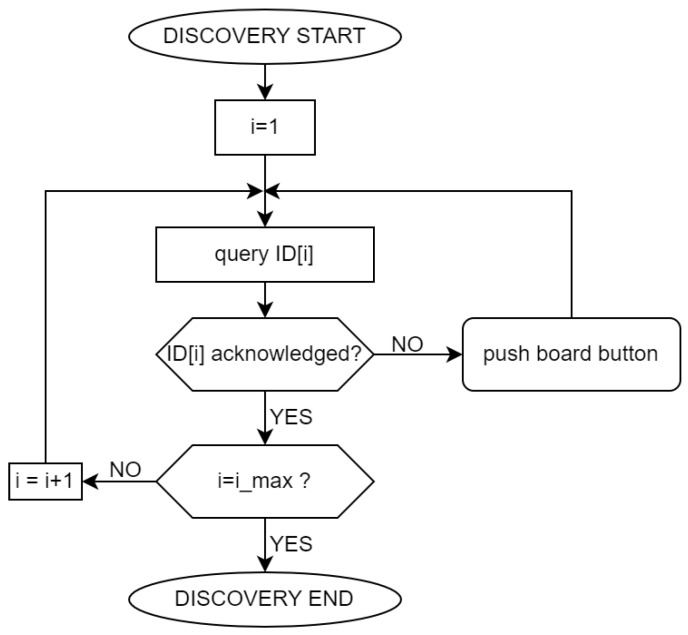
Flowchart diagram illustrating the discovery procedure that maps tiles (1 to i_*max*_) as required by the game controller.

**Figure 7 sensors-23-03162-f007:**
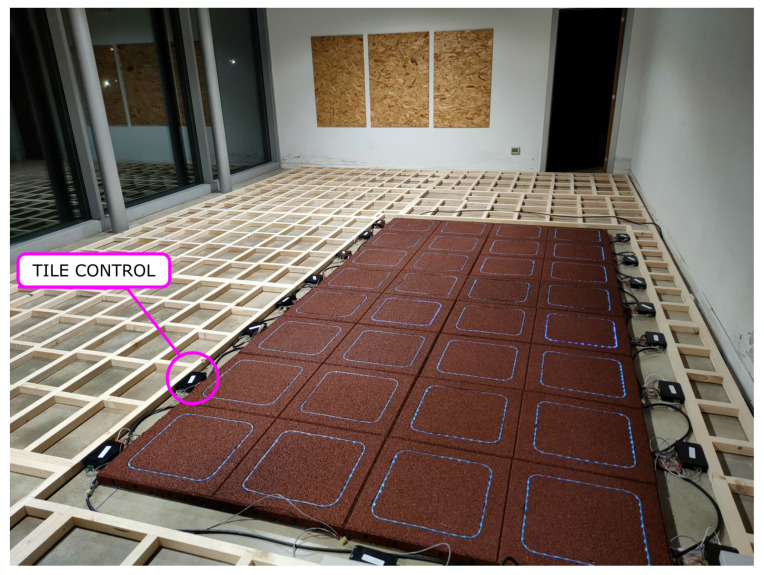
View of the demonstrator site assembly.

**Figure 8 sensors-23-03162-f008:**
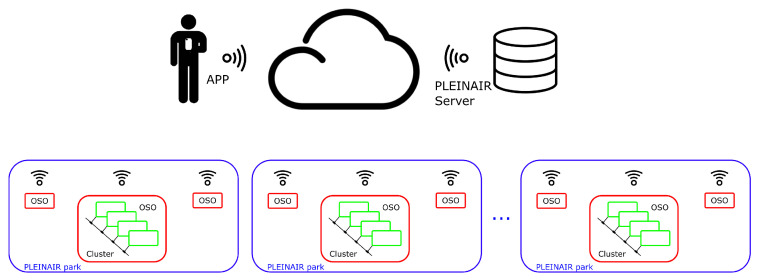
General view of the PLEINAIR system architecture.

**Figure 9 sensors-23-03162-f009:**
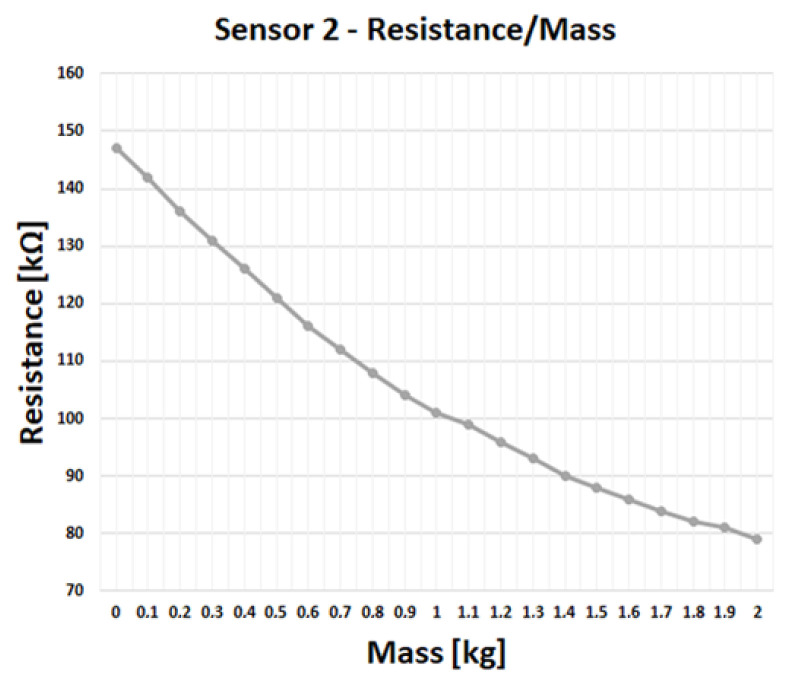
Measured resistance depending on the applied mass, for a given sensor position.

**Figure 10 sensors-23-03162-f010:**
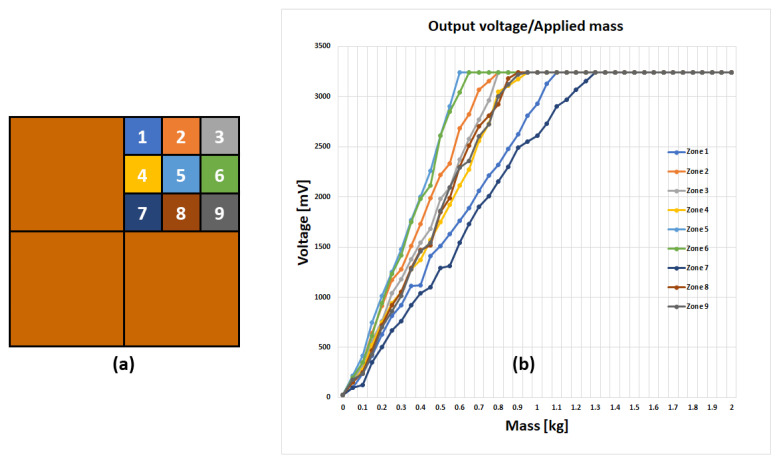
Representation of the smart tile with one quarter subdivided into nine different zones (**a**). Measures of the analogue output voltage, depending on mass and position (**b**).

**Figure 11 sensors-23-03162-f011:**
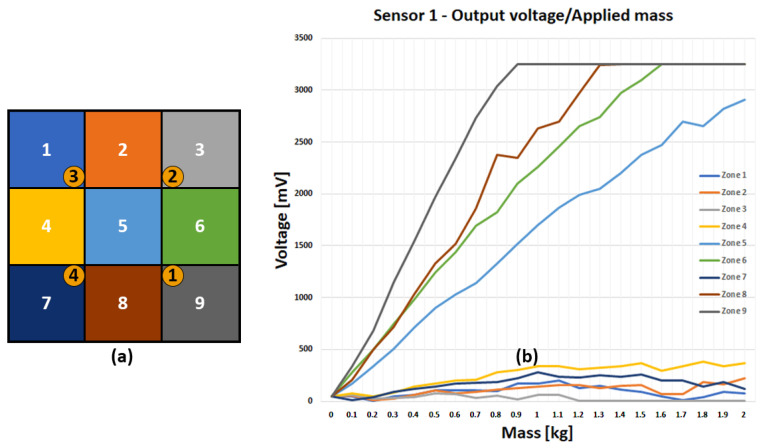
Representation of the smart tile, subdivided into nine zones (**a**). Measures of the analogue output voltage depending on mass and position (**b**).

**Figure 12 sensors-23-03162-f012:**
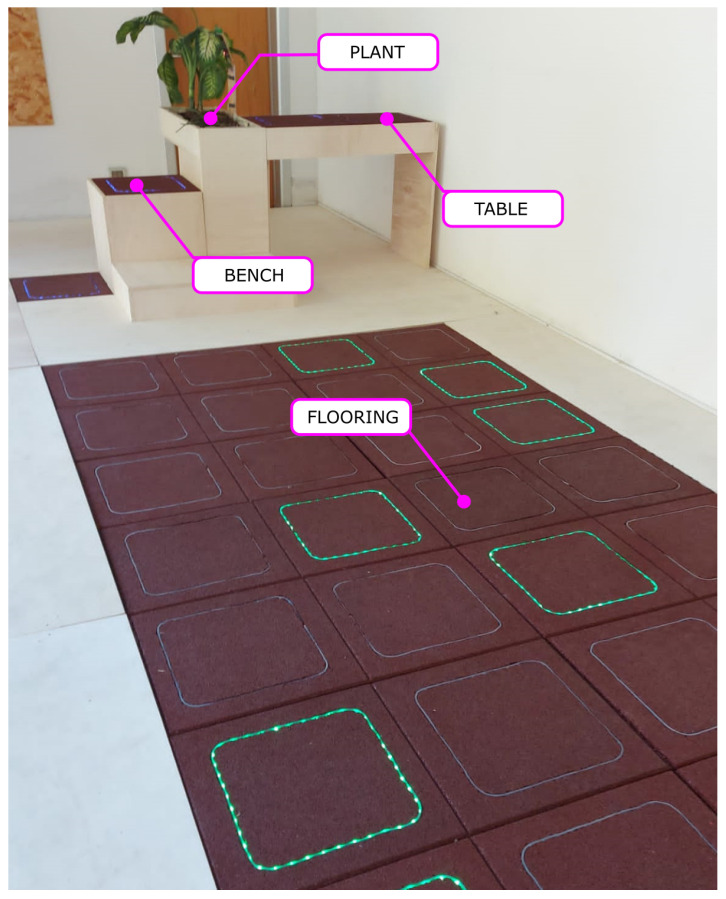
IoT OSOs installed at the “Museo della Civiltà Contadina”.

**Figure 13 sensors-23-03162-f013:**
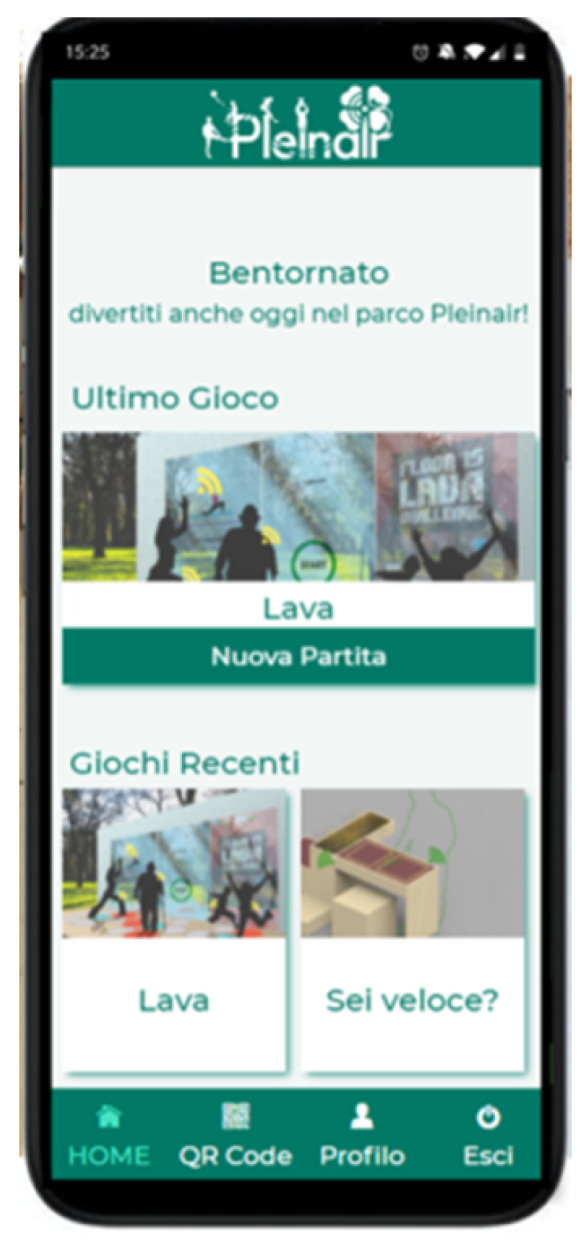
Web application screenshot.

**Figure 14 sensors-23-03162-f014:**
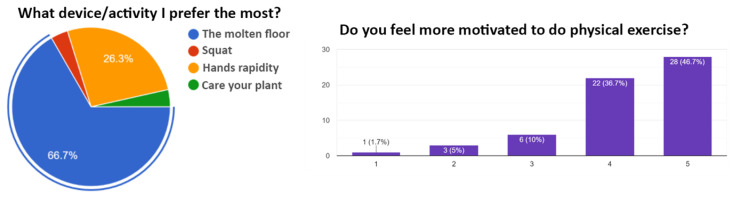
A sample of the user satisfaction assessment.

## Data Availability

Not applicable.

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
