# Peer review of "IoT Smart Flooring Supporting Active and Healthy Lifestyles"

_sensors, 2023, doi:10.3390/s23063162_

Round 1

Reviewer 1 Report

The authors present the design and implementation of a prominent demonstrator of the OSO concept, consisting of a smart,  sensitive flooring, based on anti-trauma floors commonly found in kids' playgrounds

This a very nice and easy-to-read paper on a topic very suitable for this journal. My main concerts are listed below:

1- Will be possible to provide a comparison between similar works?

2 - What are the overheads (e.g., system setup, data storage, computing efficiency, etc.) of the proposed scheme? More analysis should be provided.

3- Some references are bad formatted (line 369)

4- More technical papers about IoT using the same protocols

[1] Internet of Things. In Manual of Digital Earth (pp. 387-423). Springer, Singapore.

[2] Development of an open sensorized platform in a smart agriculture context: A vineyard support system for monitoring mildew disease. Sustainable Computing: Informatics and Systems 28, 100309

Author Response

Dear Reviewer, with reference to the paper “IoT smart-flooring supporting active and healthy lifestyles”, by Federico Cocconcelli, Guido Matrella, Niccolò Mora, Ion Casu, David Alejandro Vargas Godoy and Paolo Ciampolini, submitted for publication on MDPI Sensors, section “Internet of Things”, special issue “Sensing Technologies and IoT for Ambient Assisted Living”, please find attached our answers to your kind requests and comments.

Reviewer 2 Report

This paper is about the implementation of Smart Pad (Flooring pad) implemented as the concept of Outdoor Smart Object that can encourage physical exercise reflecting the characteristics of the game and induce interest.

However, the paper description form is in the form of a technical report that focuses on implementation rather than a paper format that requires originality.

If it is a research paper, it should clearly describe the problem to be solved, how other researchers have solved the problem, and how the authors approached and solved the problem, and what the result was. do.

The description method of this paper focused on the description of the implementation method combining existing technologies, such as how the implementation was done, which electronic devices or sensors were used, and which communication infrastructure was adopted.

An attempt was made to solve a certain problem, and it was difficult to find any mention or analysis of the originality or specificity of the solution.

The results of the experiments (Figures 8 to 10) seem to be described as the results of measuring the characteristics of the Tekscan sensors used by the authors, so it is difficult to judge this paper as the realization of an original idea. This is because this sensor is a very commonly used sensor for implementing foot pressure pads.

It can be claimed that a new and original smart flooring device has been developed, but it is unlikely that a new concept IoT device that applies the original technology claimed by the authors has been realized.

Author Response

(The authors gave the same response as above.)

Round 2

Reviewer 1 Report

The suggestions/comments have been considered. The manuscript has improved remarkably

Author Response

Thanks for your careful review and for valuable suggestions. 

Reviewer 2 Report

This paper is a paper at the implementation level and cannot mention creativity or originality, which are the most important values as a paper.

However, to this point, the authors give a somewhat erratic answer as follows.

"we believe the originality of the paper stems from the specific application, which is, to our knowledge, unprecedented. We did not claim for introducing any new IOT concepts, but instead we discuss the exploitation of the IoT paradigm to a new application field.."

This answer is judged to be lacking in the basic qualifications for writing a technical paper, and it is an unacceptable statement as a reviewer of technical paper. At least, if the authors have applied existing techniques to new applications, they should be able to derive something new from them. But I couldn't find out what the new one was.

Author Response

We fully respect your point of view, and thank you for your review.  But, at the same time, we regardfully disagree with it.   

Our paper describes the full development path of a genuinely innovative application, based on IOT technologies, starting from motivation and contextualization and discussing all steps of design, development, deployment and end-user test. To our knowledge, there is no similar approach in the literature, so far. We believe this could be of some interest to the research community and could well fit the scope of the special issue “Sensing Technologies and IoT for Ambient Assisted Living” this paper is aimed to.  

We casted the paper in the form we thought to be more appropriate for such intended audience and which is common to many respected papers in the AAL and health-related application field.   We shall be glad if the paper, in its present form,  will be eventually considered worth publishing: otherwise,  the Editor might rightfully consider it not suitable for publication.